# Behavioral Economics in the Epidemiology of the COVID-19 Pandemic: Theory and Simulations

**DOI:** 10.3390/ijerph19159557

**Published:** 2022-08-03

**Authors:** Blas A. Marin-Lopez, David Jimenez-Gomez, José-María Abellán-Perpiñán

**Affiliations:** 1Fundamentos del Análisis Económico (FAE), University of Alicante, 03690 Alicante, Spain; davidjimenezgomez@ua.es; 2Applied Economics Department, University of Murcia, 30100 Murcia, Spain; dionisos@um.es

**Keywords:** COVID-19, epidemiology, game theory, behavioral economics, public health policy

## Abstract

We provide a game-theoretical epidemiological model for the COVID-19 pandemic that takes into account that: (1) asymptomatic individuals can be contagious, (2) contagion is behavior-dependent, (3) behavior is determined by a game that depends on beliefs and social interactions, (4) there can be systematic biases in the perceptions and beliefs about the pandemic. We incorporate lockdown decisions by the government into the model. The citizens’ and government’s beliefs can exhibit several biases that we discuss from the point of view of behavioral economics. We provide simulations to understand the effect of lockdown decisions and the possibility of “nudging” citizens in the right direction by improving the accuracy of their beliefs.

## 1. Introduction

The 2019 coronavirus disease outbreak (COVID-19) has posed a significant threat to human health and the global economy, resulting in a global public health emergency. Access to approved treatments and vaccines for this disease had only recently become available when writing this article. That means that for several months during the pandemic, governments and organizations have had to rely on preventive measures such as lockdowns, social distancing, and hygienic behaviors, e.g., the use of face masks and hand-washing. As a result, a clear goal of public health policy during this outbreak has been to encourage people to practice responsible preventive behavior to limit the spread of COVID-19. Behavioral economics can help us understand how to design effective policies, implement them, and who the target population should be. This field applies insights from psychology and neuroscience to explain why people deviate from rational choice theory due to biases people may face during their decision-making process. Various scholars have pointed out that identifying these behavioral biases in the population can help predict how public health policies will be received and thus prevent them from backfiring in the current global situation [1,2,3]. Indeed, behavioral economics has been shown to produce highly effective and low-cost policies on health-related issues [4,5,6]. In addition, several articles have echoed the importance of policymakers taking into account individuals’ cognitive biases, as this allows for more efficient policy design, including policies related to public health [3,7,8].

We incorporate these insights of behavioral economics into a formal epidemiological model (SEIR-like). Our research hypothesis is that including those insights would change the predictions of SEIR-like models in relevant ways. In particular, we expect that “behavioral interventions” (that change the individual’s perceptions of the environment) can mitigate contagion. Therefore, such epidemiological models incorporating behavioral economics can be used to study how the dynamics of the pandemic (i.e., people infected and recovered) may change due to both individual and government behavior and their interaction as people’s behavior determines to some extent, the severity of the pandemic, and thus its economic consequences. We are the first, to the best of our knowledge, to combine epidemiological, game-theoretical, and behavioral assumptions to predict the progression of a pandemic. Indeed, ref. [9] argues that cooperation between economists and epidemiologists is required to tackle pandemics because it is a complex, cross-disciplinary issue that affects all aspects of society.

We build on two different kinds of literature. The first combines economics and epidemiology. Examples of this strand of literature are the studies on endogenizing economic behavior in a rational framework [10]; considering that the rate at which individuals change their sexual partner is endogenously determined by the prevalence of HIV [11]; a rational choice framework in which individuals decide on several aspects of their sexual relationships [12]; and allowing for forward-looking risky behavior [13]. More recently, this type of model has been applied to the study of COVID-19 [14,15,16]. The second literature incorporates game theory into epidemiological models, for example, considering myopic agents who can choose a continuous variable of protection [17]; and, more recently, addressing variants of the SIR model in which individuals reduce contacts as a function of the current or cumulative number of cases [18].

However, to the best of our knowledge, there has been no attempt to incorporate behavioral economics into a game-theoretical epidemiological model. This means that the analysis either comes from behavioral economics but with no formal epidemiological models or from epidemiological models that do not consider behavioral economics. In the present article, we aim to integrate behavioral economics into a game-theoretical epidemiological model with the goals of improving the predictions of the model and to better formalize some of the arguments that have been advanced from behavioral economics about the management of COVID-19.

## 2. Materials and Methods

### 2.1. SEIR Model, Game Theory, and Behavioral Economics

In this paper, we build upon the well-known compartmental epidemiological models as Susceptible-Exposed-Infectious-Removed (SEIR) that have been useful in studying pandemic dynamics. These models demonstrate how various public health strategies can influence the epidemic’s outcome. However, we believe that considering how behavioral biases can influence pandemic dynamics through individual actions can yield a wealth of information. Concerning this, we are the first, to our knowledge, to combine game theory and behavioral economics with an epidemiological model. This allows us to (1) endogenize the behavior of the population and, in particular, to allow agents to react to incentives and their perceptions of the progression of the pandemic; (2) to formalize how cognitive biases affect people’s perceptions, which in turn affect their behavior and, ultimately, the dynamics of contagion.

Our model has three components that, as we discussed, have never been jointly used before:1.A SEIR-type epidemiological model that, in particular, considers asymptomatic transmission for the case of COVID-19.2.A game-theoretical component in which individuals choose their level of responsible behavior (and thus the probability of contagion).3.Mechanisms from behavioral economics, considering that individuals’ perceptions and beliefs might be biased, especially as applied to COVID-19 [1,2,7,8,19]. In particular, we consider the following channels (we provide additional details about these channels in Online Appendix E).(a)**Optimism bias** and **overconfidence**, by which people tend to be excessively optimistic about future events and excessively confident in their own beliefs and skills to face those events [20].(b)**Availability heuristic**, in which the probability of an unknown event (a severe pandemic COVID-19) is estimated using the probability of a known event (a severe pandemic of seasonal influenza).(c)**Fallacy of lack of evidence**, whereby an absence of evidence for an event is interpreted as evidence for the absence of that event [21].(d)**Status quo bias**, by which people tend to prefer the current baseline (or status quo) over any change in such a way that there is a risk of doing nothing in the face of such a novel event as the pandemic [22].(e)**Social influences** have been extensively studied as a critical element in their effect on people’s choices and behaviors [23], especially on health-related outcomes.

### 2.2. Calibration

The simulations are all 360 periods long, e.g., t=1,⋯,360, except those for the extended model, where we use 180 periods in the interest of computational time (included in the Online Appendix B). (However, this does not affect the shape of the dynamics, e.g., in terms of the evolution of cases, and therefore we can still compare the results across simulations.) To solve the system of non-linear differential equations, we have used a standard Runge-Kutta algorithm for the integration. Moreover, the parameters included in the model can be divided into two groups: medical-epidemiological parameters and behavioral parameters. The first group includes the subclinical infectious rate Asymptomatic vs. Infected, α; the (baseline) transmission rate, β; the recovery rate, γ; the proportion of asymptomatic/subclinical cases, ρ; the mortality rate, δ; and the incubation rate, κ. These parameters are summarized in Table 1, where the value used in the simulations and its corresponding reference can be seen. Some of these values are directly given by the SARS-CoV-2 characteristics themselves, such as the recovery rate (the inverse of the days needed for full recovery) and the incubation rate (the inverse of the days needed to incubate the virus). However, the remaining values are taken from the existing literature, which has estimated them using epidemiological models.

The set of behavioral parameters that are included in the model are summarized in Table 2, where the theoretical value in the model, the value used in the simulations, and a summary of its interpretation from a behavioral economics point of view are presented.

## 3. Theory

### 3.1. SEAIRD Baseline Model

We describe a discrete-time SEAIRD model of COVID-19: Susceptible, Exposed, Asymptomatic (subclinical) infected, symptomatic (clinical) Infected, Recovered, and Deceased. (The SEAIRD model is an SEIR model with an additional category for infected but asymptomatic and another for deceased. Those in both *E* and *A* are unaware of their status. The implication of the model is that both exposed and asymptomatic individuals will behave as susceptible. However, asymptomatic infected individuals will have an infectious potential). We build our model following [30] (note that all variables depend implicitly on time *t*):(1)$$\begin{array}{cc} \Delta S & =-S[\beta_{A}\left(a\right)A+\beta_{I}\left(a\right)I], \end{array}$$(2)$$\begin{array}{cc} \Delta E & =-\kappa E+S[\beta_{A}\left(a\right)A+\beta_{I}\left(a\right)I], \end{array}$$(3)$$\begin{array}{cc} \Delta A & =-\gamma A+\rho \kappa E, \end{array}$$(4)$$\begin{array}{cc} \Delta I & =-\gamma I+(1-\rho )\kappa E, \end{array}$$(5)$$\begin{array}{cc} \Delta R & =\gamma [A+I(1-\delta )], \end{array}$$(6)$$\begin{array}{cc} \Delta D & =\delta \gamma I, \end{array}$$
where a represents the profile of actions taken by the population, which remains constant over time. Hence, we consider that the contagion rate depends on actions taken by citizens (such as protective measures) and take the rest of the medical parameters as independent from behavior. Following other models that consider asymptomatic individuals, e.g., [31], we consider that those in *A* and *I* have different infectious potential and behavior. In terms of the dynamics of the disease, successful individuals first become exposed with a certain probability that, as we have mentioned, depends on the behavior of asymptomatic and symptomatic infected individuals, which we explained below in more detail. (There is not a single “beta” parameter, but rather two: $\beta_{A}\left(a\right)$ and $\beta_{I}\left(a\right)$, and they depend on the individual’s behavior. Indeed, it is precisely this dependence on behavior for the two categories that generate the need for two such parameters.) Those who are exposed will develop the disease with probability κ, and of those, they will either become infected asymptomatic (with probability ρ) or infected symptomatic (with probability 1−ρ). Finally, individuals who are infected (either symptomatic or asymptomatic) will recover with probability γ, except for a fraction δ of those, who will die of the disease.

### 3.2. Individual Behavior

At each period of time, individuals decide whether to engage in responsible behavior or not. This responsible behavior (that we label “protect”) can include taking protective measures such as wearing a face mask, washing hands regularly, keeping social distance, etc. (Deceased individuals do not choose any action, and so from now on when we discuss behavior, we will refer exclusively to each of the other categories.) Moreover, individuals have a belief $p_{C}$ about the probability of the disease being capable of causing a pandemic. We argue that the several biases that we considered in Section 2, imply that $p_{C}<p$, i.e., the individual undervalues the probability of a severe pandemic (in Online Appendix E we provide a detailed account about how these biases would affect $p_{C}$). This parameter allows us to capture several cognitive biases that could potentially happen at the beginning of the pandemic when individuals might underestimate the probability of the virus generating a pandemic. In addition, each individual has beliefs $\hat{\beta }\left(a\right)$ about the contagion that takes place (depending on the action she takes). While we give the explicit definition below, $\hat{\beta }\left(a\right)$ should be interpreted as the probability that an interaction between an individual who chooses *a* and another random individual in the population will result in contagion.

Individuals suffer a disutility −D<0 from contracting the disease, and each individual has a private cost *c* of protecting oneself, normally distributed $N(\mu_{c},\sigma_{c})$. Moreover, we assume that the psychological benefit or cost from protecting oneself $S(\bar{a})$ depends only on the average action $\bar{a}$ taken by the population. Function $S(\bar{a})$ could be negative (i.e., at the beginning of the pandemic, when few people are engaging in protective measures, they can be ridiculed for doing so) or positive. We assume that those infected always have an extra “moral” utility $S^{I}>0$ when engaging in responsible behavior. Thus, the payoffs for taking a certain action depend on the individual’s status, as described in the table below (Table 3).

There are several important details to consider. First, we follow the epidemiological literature that incorporates game theory, such as [17], assuming that individuals are not forward-looking but rather that they best respond to their incentives each period. Second, since we assumed that individuals in the categories S,E, and *A* cannot differentiate between them (that is, individuals are unable to identify whether they are susceptible, exposed, or infected but asymptomatic), the incentives for individuals in those categories are identical. Note also that the government can influence the citizens’ perceptions, for example, by making them aware of the severity of the pandemic (hence making $p_{C}$ more accurate) or by using nudges that influence their perception $\hat{\beta }\left(a\right)$ over the contagion rate, something we discuss in Section 4.2.3.

As we noted in the Introduction, our model departs from many epidemiological models in having two distinct functions $\beta_{A}\left(a\right)$, $\beta_{I}\left(a\right)$, the rate at which susceptible agents become infected by asymptomatic and infected individuals, respectively. These rates are derived by assuming that each individual interacts randomly with another individual each period. Following [30], we assume that parameter β measures the baseline contagion rate from those in *I* when there is no lockdown and no protective measures are taken. In contrast, for those in *A*, this rate is multiplied by a factor α<1, willing to capture the biological difference in the contagion rate between asymptomatic and symptomatic individuals. Moreover, we assume that for any infectious individual (symptomatic or not), their contagion is reduced by a factor of ν when they use protective measures and by a factor ${ℵ}_{L}$ that depends on whether lockdown is in effect (L=1, a case that we discuss below in Section 3.3), or not L=0. (In particular, if there is no lockdown L=0, then ${ℵ}_{0}=1$. If there is a lockdown, then ${ℵ}_{1}$ is an exogenous parameter determining how effective the lockdown is in reducing contagion when agents respect the lockdown. Ref. [32] consider instead the *effectiveness of lockdown*
λ. Therefore, ${ℵ}_{1}=1-\lambda$.) With these assumptions in place, we have that $\beta_{A}\left(a\right)$ and $\beta_{I}\left(a\right)$ are given by:(7)$$\begin{array}{c} \beta_{A}\left(a\right)=\alpha \beta \cdot E\left[\left(1-\nu {ℵ}_{L}a_{S}\right)\left(1-\nu {ℵ}_{L}a_{A}\right)\right], \end{array}$$(8)$$\begin{array}{c} \beta_{I}\left(a\right)=\beta \cdot E\left[\left(1-\nu {ℵ}_{L}a_{S}\right)\left(1-\nu {ℵ}_{L}a_{I}\right)\right], \end{array}$$
where the expectation is taken over behavior in each compartment (i.e., $a_{S},a_{A},a_{I}$), as the cost of choosing protective measures is idiosyncratic for each individual. Note that for a susceptible individual, the relevant rate is given by β(a), that is a weighted sum of $\beta_{A}\left(a\right)$ and $\beta_{I}\left(a\right)$, conditional on the action *a* chosen by the individual
$$\beta \left(a\right)=\beta \left(1-\nu {ℵ}_{L}a\right)E\left[\alpha A\left(1-\nu {ℵ}_{L}a_{A}\right)+I\left(1-\nu {ℵ}_{L}a_{I}\right)\right].$$

However, we allow individuals to have a biased belief about ν, which we denote by $\hat{\nu }$, and therefore they might have a biased belief about β(a), which we denote $\hat{\beta }\left(a\right)$. ($\hat{\beta }\left(a\right)$ would be defined as $\hat{\beta }\left(a\right)=\beta \left(1-\hat{\nu }{ℵ}_{L}a\right)E\left[\alpha A\left(1-\hat{\nu }{ℵ}_{L}a_{A}\right)+I\left(1-\hat{\nu }{ℵ}_{L}a_{I}\right)\right].$ In other words, individuals understand the basic dynamics of the contagion rates when they compute the utility of choosing actions, but we allow for the possibility that they are biased in their estimation of the effectiveness of responsible behavior ν and, therefore, believe that it is given by $\hat{\nu }$. Obviously, if the individuals held correct beliefs about ν, then $\hat{\beta }\left(a\right)=\beta \left(a\right)$.)

#### Solving the Baseline Model with No Lockdown

From the game payoffs it follows immediately that an individual with cost *c* uses protective measures whenever $S\left(\bar{a}\right)+S^{I}>c$ if she is in *I*, and whenever $S(\bar{a})>c$ if she is in *R*. Pre-symptomatic individuals (in S,E,A) with cost *c* use protective measures whenever
$$S\left(\bar{a}\right)-p_{C}\hat{\beta }\left(1\right)D-c>-p_{C}\hat{\beta }\left(0\right)D\;⟺\;S\left(\bar{a}\right)>c-p_{C}D\left[\hat{\beta }\left(0\right)-\hat{\beta }\left(1\right)\right].$$

Therefore, for each category of individuals X∈{S,E,A,I,R} there is a threshold $c_{X}$ (with $c_{S}=c_{E}=c_{A}$), such that the individual protects herself if her cost is below the threshold. Hence, the fraction of individuals in *S* who protect themselves is $\Phi \left(\frac{c_{S}-\mu_{c}}{\sigma_{c}}\right)$ (where Φ is the standard normal CDF), and analogously for the other four categories. Note that at the beginning of the pandemic, we have A≈I≈0, and, therefore, the incentives for susceptible individuals are approximately the same as for recovered individuals, namely those given by social norms.

### 3.3. Lockdown

Having mentioned the possibility of lockdown in the previous section, we now consider how it might occur. In our model, we assume that the government is the only player who can choose the lockdown. The government derives a disutility *q* from lockdown (as it reduces economic activity or increases criticism) per period that the lockdown is enforced. However, the government also derives a disutility from the deaths during the pandemic. Therefore, the expected utility of the government (we assume no time-discounting at this stage) at time *T* when a successful vaccination campaign is implemented and the pandemic ends is:(9)$$V_{y}^{g}=-\sum_{t=0}^{T}\left\{qQ_{t}+p_{G}D_{t}\right\},$$
where $p_{G}$ is the probability the government assigns to the pandemic, and $Q_{t}=1$ if there was a lockdown at time *t*, and 0 otherwise. As we assume that deaths can only be produced by the pandemic, we then have that $E\left[D_{t}\right]=p_{G}D_{t}+\left(1-p_{G}\right)\cdot 0=p_{G}D_{t}$.

**Assumption** **A1.**
*The lockdown enacted by the government has the following restrictions: (1) the lockdown can only be approved in discrete segments of duration τ; (2) once a lockdown is not in place, the government must wait for a period τ to begin a new lockdown; and (3) the first time a decision about lockdown must be made is at time t=0.*


In practice, Assumption 1 means that the government can only enact lockdown at time t=0,τ,2τ,⋯, until time *T* when a vaccine is discovered and implemented (for simplicity, we abstract away from considering the issues associated with implementing the vaccine). (In Online Appendix B, we consider an extension with a political economy model that allows for two types of government (authoritarian and democratic), as well as the possibility of declaring the state of alarm. We then explore the interaction between beliefs about the pandemic and the government’s response.)

## 4. Results

### 4.1. Theoretical Results

Asymptomatic individuals are biologically less contagious than symptomatic Infected individuals (because of α<1 in Equation (Equation 7)). However, they can be more contagious once we take into account their behavior. Let ${\bar{a}}_{I},{\bar{a}}_{A}$ be the average action taken by Infected and Asymptomatic individuals, respectively.

**Proposition** **1.**
*Let $\alpha^{*}\left(\nu ,{ℵ}_{L},{\bar{a}}_{I},{\bar{a}}_{A}\right)$ be defined as*

(10)
$$\alpha^{*}=\frac{1-\nu {ℵ}_{L}{\bar{a}}_{I}}{1-\nu {ℵ}_{L}{\bar{a}}_{A}}.$$


*Then, whenever $\alpha >\alpha^{*}$, Asymptomatic individuals are more contagious than symptomatic Infected individuals.*


While Asymptomatic individuals are *biologically* less infectious, Proposition 1 highlights the case when they are *behaviorally* more contagious than infected individuals, as they are not aware of transmitting the disease. Therefore, we have two opposing forces. In our model, when $\alpha >\alpha^{*}$, the behavioral contagion channel is stronger than the biological channel, and that is why Asymptomatic are more contagious than symptomatic Infected individuals. This is a crucial insight to understanding the transmission of SARS-CoV-2: as asymptomatic individuals may not take precautions (e.g., by wearing face masks, washing hands), they become more contagious than symptomatic individuals, and they are then more likely to transmit the virus to the susceptible population.

**Proposition** **2.**
*The threshold $\alpha^{*}$ is increasing in $\hat{\nu }$. A sufficient condition for $\alpha^{*}$ to be increasing in ν and ${ℵ}_{L}$ is that ${\bar{a}}_{A}>{\bar{a}}_{I}$.*


Proposition 2 shows that when the citizen’s perception $\hat{\nu }$ of the efficacy of protective measures increases, it becomes more difficult for Asymptomatic individuals to be more contagious than symptomatic Infected individuals. The intuition for this is that when $\hat{\nu }$ increases, Asymptomatic individuals (who behave as Susceptible, as they are not aware of being infected) are more willing to engage in protective behavior, which reduces their infectiousness. On the other hand, when the actual effectiveness of protective measures ν or lockdown ${ℵ}_{L}$ increase, the effect on $\alpha^{*}$ is a priori ambiguous; however, when ${\bar{a}}_{A}>{\bar{a}}_{I}$, we can say with certainty that $\alpha^{*}$ is increasing. Therefore, beyond the efforts that governments can make to improve the actual effectiveness of protective measures, it is essential to note that increasing citizens’ perceptions another helpful strategy for fighting COVID-19, something we discuss further in the next section. (We perform similar simulations for the efficacy of lockdown measures in Online Appendix D).

### 4.2. Numerical Results

In this section, we present the results of the model simulations for each of the six categories contained in the SEAIRD behavioral epidemiology model. We want to stress the fact that there is great uncertainty about the key parameters for the COVID-19 epidemic [33]. This uncertainty extends to the design of an optimal policy, as these parameters are extremely sensitive to changes [34,35,36]. Moreover, simulations are often performed under strong assumptions about the impact of social distance policies without connecting to the necessary data [37]. As a result, since the exact numbers in terms of economic cost and public health are affected by this [38], our analysis will focus solely on assessing the effects on the relative change in pandemic dynamics. For example, comparing the evolution of the number of infected between various scenarios rather than in absolute terms (e.g., the total number of infected). These results should then be interpreted as changes in the dynamics (evolution) between categories. Concerning this, any policy that has focused on containing the spread of the virus has aimed to “flatten the curve”, and thus reduce the rate at which society as a whole goes from Susceptible to Exposed, that is, the contagion rate. Based on the model and the results presented below, any policymaker should consider several factors that may influence the dynamics of the pandemic. In this regard, two factors that have a major effect on this evolution and have a direct link to behavioral economics are (1) the effectiveness of preventive measures ν (as well as the perception that individuals have of them, $\hat{\nu }$), and (2) the social norms that prevail in the country or “country” norms. In the following subsections, we will further analyze the two factors mentioned above.

#### 4.2.1. Effectiveness of the Use of Protective Measures

As we can see in Figure 1, an increase in efficacy ν translates into a slowing down of the outbreak. (Other factors that affect the time-course of the outbreak. The graph for the efficacy of lockdown ${ℵ}_{L=1}$ is similar to that in Figure 1 and can be found in the Online Appendix D. These parameters directly affect the contagion rate of A ($\beta_{A}$), and I ($\beta_{I}$), decreasing the rate at which individuals in *S* flow to the *E* category.) That is to say, as the efficacy increases, the number of infected cases becomes more homogeneously distributed during the period of time considered in the simulation. In other words, a public health policy aimed at increasing the effectiveness of the use of protective measures will produce a flattening of the curve in terms of the proportion of cases. Examples of such policies could be an introduction of mandatory face mask use, the use of hydroalcoholic gel in establishments, or a policy that establishes social distance.

#### 4.2.2. Social Norms

Social norms have been emphasized as one of the key factors in determining behavior during the pandemic, and, therefore, effective rates of contagion [3,7,39,40,41,42,43,44,45]. We have modeled social norms by including $S(\bar{a})$ as a payoff for engaging in protective measures for all individuals (symptomatic individuals derive an extra $S^{I}>0$). For the sake of deriving numerical results, we model function $S\left(\bar{a}\right)=\pi \left(\bar{a}\right)+\varsigma$, where $\pi (\bar{a})$ is a sigmoid function, and ς is a constant that is intended to capture “shifts” in the importance of social norms across differences societies (thus, ς is also a “behavioral” parameter).

As we can see in Figure 2, an increase of ς has the expected effect on the evolution of the number of cases in terms of flattening the curve over the time period. Higher values of ς reflect stronger norms (possibly country-specific), which translates into a higher proportion of individuals being protected from the onset of the pandemic. For instance, in the case of face masks, those countries that started their widespread use (mask-wearing culture) have been able to contain the virus better in terms of the rate of contagion. This is because public mask-wearing is most effective at preventing virus spread when compliance is high [46]. Concerning this, in the absence of widespread vaccination for COVID-19, governments and public health institutions have ended up being advocated for individuals to use preventive measures during the pandemic. Therefore, developing public health policies that affect (in this case, social norms) is vital to slow down the time-course of the outbreak. In addition, policymakers must also take into account potential free-riding behavior. In the case of face masks, a greater use of them in the community may encourage others to follow the norm. However, it also creates incentives for others not to assume the cost of protecting themselves since others already do so [45].

Additionally, it is different to start at an early stage of the pandemic in a country with a culture of preventive measures from another society that promotes this social norm in more advanced pandemic stages. Thus, in the following simulation, we exploit the timing of a policy aimed at shifting social norms in two different periods. We can see in Figure 3 that the model captures well this “timing effect” or anticipation in the public policies. This is of great importance since nudges that directly affect social norms by inducing a change in behavior at the beginning of the pandemic have a more significant effect at “flattening the curve” than when it is at a more advanced stage.

#### 4.2.3. Mandates vs. Nudges to Fight COVID

Individuals’ decisions to use preventive measures are greatly affected by both their beliefs about their efficacy as well as the prevalence of their behavior, e.g., the use of face masks in public [45]. As part of these preventive measures, wearing a mask is highly effective. It may protect both the user and those around them from contracting COVID-19 [40,46,47,48,49,50]. Moreover, the effectiveness of other measures, such as hand-washing or social distancing, is also decisive in influencing the number of reported daily cases, for instance, in Thailand, ref. [51]. Thus, the effectiveness of any preventive measures against the transmission of the virus can be very high. However, if citizens believe that it is not effective, that will guide their beliefs and, therefore, their actions (they might fail to protect themselves). Instead of trying to change individuals’ beliefs, the government might use mandates that increase the effectiveness of protective measures, corresponding to an increase in ν in our model (imposing mandatory use of masks, use of hydroalcoholic gel in establishments, social distancing, etc.), as discussed in Section 4.2.1. (In our model, mandates have a qualitatively similar effect to a change in social norms, leading to a flattening of the curve.)

In terms of changing not only the actual effectiveness ν, but also the perception about it $\hat{\nu }$, a policy maker might at first think that both would have the same effect on the evolution of cases (although not in proportion). However, the intuition is not straightforward. In our model, a shift in perception $\hat{\nu }$ has minimal effect on the dynamics of the pandemic, which is striking because it is the perceptions that guide, to some extent, individuals’ behavior. To better understand this phenomenon, note that at the beginning of the pandemic, the number of cases is so small that a change in perception does not generate sufficient incentives to move the average behavior of society (behavioral force). However, when the number of cases grows sufficiently enough so that perceptions can cause such incentives, it is “too late” because then a nudge on perception does not achieve the desired objective due to the sheer inertia of the dynamics (epidemiological force). Indeed, ref. [52] find using a causal structural model that in the US, a nationwide mandate on the use of face masks for employees in the early stages of the pandemic could have resulted in a reduction of weekly case growth by more than ten percentage points in late April.

About a mandate forcing individuals to wear face masks in the early stages of the pandemic versus later stages, we find the same timing effect as with social norms. (As was the case with social norms, the timing of behavioral policies also matters. In the Online Appendix D, we include an exercise similar to that in Figure 3 and find a similar result: implementing the nudge earlier is more effective in flattening the curve.) This emphasizes the importance of taking direct and unhesitating action even when the number of cases is low, as the number of cases is growing exponentially. As [53] noted, there are two main reasons why countries like Taiwan have been able to control the virus much more quickly: first, Taiwan had a robust public health infrastructure in place before COVID-19, allowing for faster and more coordinated responses; and second, widespread use of face masks to reduce transmission from infected people (regardless of symptoms) while also protecting wearers (mass wearing). This leads us to propose the exercise of comparing two scenarios: one in which a mask-wearing culture with citizens with greater accuracy in their beliefs about the actual effectiveness of protective measures, against the opposite scenario. As we can see in Figure 4, a country with higher social norms and more accurate beliefs about the effectiveness of protection can flatten the curve, reducing then the incidence of the pandemic.

## 5. Conclusions

As stated in the Introduction, our research hypothesis was that introducing behavioral insights and game theory into an SEIR-like epidemiological model would change the predictions of those models in relevant ways. We have validated this hypothesis by providing a series of novel results. We provide here a summary of those results.

A relevant finding is that, even if asymptomatic infected individuals are biologically less contagious than symptomatic individuals, for example, due to a lower virus load, ref. [54], they can be effectively more contagious (Proposition 1), the reason being that those with no symptoms might not take preventive measures against COVID-19 with enough intensity. However, we show that this is less likely to happen if the perceived effectiveness of protecting oneself increases (Proposition 2), as this in turn increases the perceived incentives for asymptomatic individuals to engage in protective behavior, thus reducing their infectiousness and lowering contagion. Theoretically, this means that governments can then mitigate the pandemic not only through lockdowns or by making it mandatory to wear face masks but also by “nudging” people to protect themselves and the rest of society [3].

Concerning our numerical results, we show that an increase in the effectiveness of protective measures “flattens the curve” of contagion, as is to be expected. However, an interesting result is that, perhaps surprisingly, improving citizen’s beliefs about the effectiveness of protective measures has a small effect on the pandemic dynamics. This is because at the start of the pandemic, improving citizens’ perceptions about protective measures has a small effect because there are few cases (and thus individual incentives are small), and by the time there are enough cases, it is “too late” to affect the dynamics of contagion meaningfully. The same is true for social norms which, by themselves, might not be enough to stop the spread of the pandemic in countries in which the initial social norm is not to engage in protective behavior, for a similar reason as the one discussed for nudges about preventive measures effectiveness. For these reasons, mandates might be more effective at the beginning of the pandemic, in terms of reducing contagion, than nudges or a sudden increase in social pressure. (However, as we argue in [55], nudges are also easier and faster to implement, and thus governments might be better off by implementing nudges in the interim while mandates are being approved.) This is one of the reasons why, in coherence with our results, mask-wearing societies with populations used to the use of protective measures have had a faster control of the pandemic [46]: the more normalized in society the use of protective measures is, the higher the probability of adopting them initially and thus when it matters most for controlling contagion and flattening the curve.

These results show that our research hypothesis was correct, and our model has produced meaningful insights that could not have been derived without considering the behavioral biases of the individuals in their decision on whether to take preventive measures. This is obviously one of the strengths of our model. However, our framework has some limitations that we would like to discuss, as well as future research directions that can solve these limitations and provide new results.

One of the limitations of our framework is that we use a “reduced form” approach, where several biases are mapped to the same behavioral parameters: a future research direction is to incorporate specific biases explicitly into the model, for example, present bias through models of hyperbolic discounting, ref. [56]. Another limitation is that the action space is small, with only two possible actions for individuals to take; an immediate avenue for future research consists of enriching the action space to include multidimensional actions (to allow for the possibility that wearing face masks may relax other protective measures), as well as heterogeneity by age and other characteristics, in such a way that the payoffs of the game depend on such characteristics. (As older people have a demonstrated higher risk of suffering the consequences of COVID-19 [57], public health policies would also affect them heterogeneously depending on age.) In addition, social networks could be included together in the epidemiological-behavioral model.

We view our paper as the first step in a research agenda that combines epidemiological models, game theory, and behavioral economics, and we believe that it opens the possibility of incorporating interesting extensions. For example, we have included a political economy model in the Appendix B, in which the government must decide how to react to the pandemic. However, there are many other economic applications that can be incorporated, among others, vaccine R&D, vaccination campaigns, etc. We view all of these as fruitful extensions that can be incorporated into our framework.

The codes needed to replicate the results obtained in the article are provided as Appendix A. In particular, they replicate all the figures included.

## Figures and Tables

**Figure 1 ijerph-19-09557-f001:**
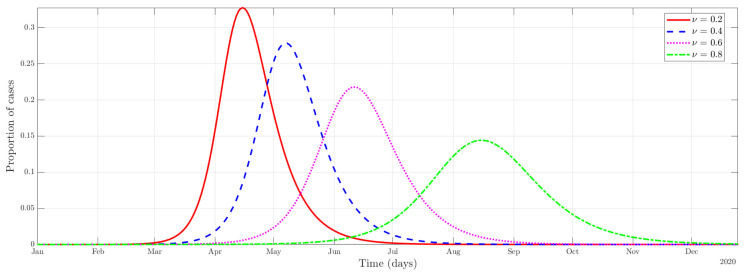
Disease dynamics of the combined infectious category (A+I) with different protective measures effectiveness.

**Figure 2 ijerph-19-09557-f002:**
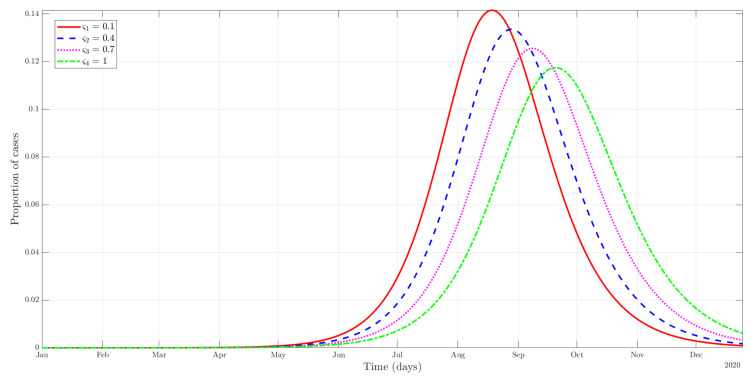
Evolution in the number of cases of the combined infectious category (A+I) depends on the strength of social norms (ς).

**Figure 3 ijerph-19-09557-f003:**
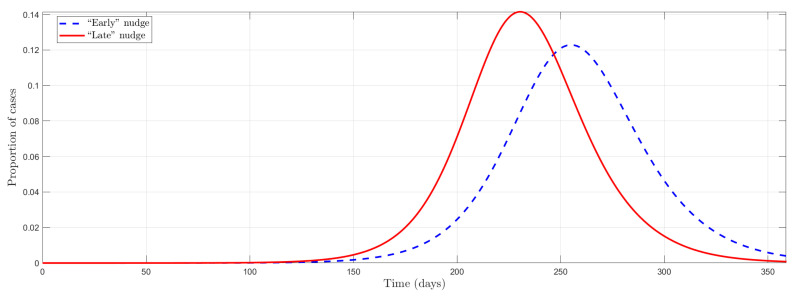
Timing effect in a nudge on social norms.

**Figure 4 ijerph-19-09557-f004:**
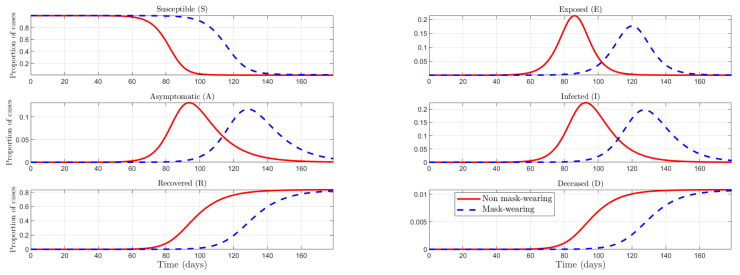
Mask wearing vs. non-mask-wearing culture comparison: the effect of social norms and efficient use of protection.

**Table 1 ijerph-19-09557-t001:** SEAIRD epidemiological parameter values.

Parameter	Value Used	Range	Description	Reference
α	0.5	-	Subclinical infectious rate A vs. I	[24]
β	0.65	0.1–1	(Baseline) Transmission rate	[25]
γ	1/14	-	Recovery rate for mild infections	[26]
ρ	1/3	0.05–0.6	Proportion of subclinical cases	[27]
δ	0.021	0.005–0.03	Death rate	[28]
κ	1/6.4	-	Incubation rate	[29]

Notes: This table shows the values of the parameters used for the model simulations. The first two parameters are taken from the epidemiology literature, being the baseline contagion rate a calibration. The last four are medically known parameters.

**Table 2 ijerph-19-09557-t002:** Relevant Behavioral and Economic parameters.

Parameter	Value Used	Theoretical Range	Description	Behavioral Interpretation
$p_{C},p_{G}$	0.5	[0,1]	Citizen’s and government’s prior	Optimism bias and overconfidence Availability heuristic Anumerism/failure to understand exponential growth Fallacy of lack of evidence Status quo bias Present bias
−D	−0.5	[−∞,0)	Disutility of contracting COVID-19	–
*c*	$\tilde{N(\mu_{c},\sigma_{c})}$	[−∞,∞]	Economic/psychological cost	–
$\mu_{c}$	0	[−∞,∞]	Population mean of the cost *c*	
$\sigma_{c}^{2}$	1	[0,∞]	Population variance of the cost *c*	
$S^{I}$	0.5	(0,∞]	Extra “moral” utility (for I)	–
${ℵ}_{L}$	${ℵ}_{0}=0.3$, ${ℵ}_{1}=1$	$\overset{{ℵ}_{L}=}{\left\{\begin{array}{cc} & {ℵ}_{1},\;\mathrm{if}\;Q=1\\ & {ℵ}_{0}\;\mathrm{otherwise} \end{array}\right.}$	Lockdown efficacy	–
ν	0.8	[0,1]	Protective measures efficacy (for S and I)	Wrong beliefs $\hat{\nu }$: Availability heuristic
ς	0	[0,∞]	Strength of social norms	Social/Country norms

Notes: These are the relevant values given to behavioral and economic parameters in the model. Each simulation in the model keeps these values constant while changing the parameter of interest analyzed.

**Table 3 ijerph-19-09557-t003:** Payoff matrix.

	Protect	Don’t Protect
S,E,A	$S\left(\bar{a}\right)-p_{C}D\hat{\beta }\left(1\right)-c$	$-p_{C}D\hat{\beta }\left(0\right)$
*I*	$S\left(\bar{a}\right)+S^{I}-c$	0
*R*	$S(\bar{a})-c$	0

## Data Availability

The authors are making the code publicly available.

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
