# Peer review of "Behavioral Economics in the Epidemiology of the COVID-19 Pandemic: Theory and Simulations"

_ijerph, 2022, doi:10.3390/ijerph19159557_

Round 1
Reviewer 1 Report
I'm not much of a specialist into game theory, but I believe your research methodology is well fundamented. I've carefully read your Appendix in order to be able to better understand your results and their discussions.
I think you should add a separate subsection of Conclusions, including the summarization of your main findings, the limits of your research and your future research directions. Please detail where do you go from here? How will you improve these?
Otherwise the manuscript is fine, it may be published once you add Conclusions as suggested before, many thanks!
Author Response
Thank you for your comment. We have deleted the Discussion section and introduced a new Conclusions section (page 11), that includes an explicit statement of our research question, as well as a summary of the theoretical and numerical results. We have also discussed the limitations of our model, as well as future research directions to improve on those limitations. In particular, we have the following paragraphs:
“These results show that our research hypothesis was correct, and our model has produced meaningful insights that could not have been derived without considering the behavioral biases of the individuals in their decision on whether to take preventive measures. This is obviously one of the strengths of our model. However, our framework has some limitations that we would like to discuss, as well as future research directions that can solve these limitations and provide new results.
One of the limitations of our framework is that we use a ``reduced form'' approach, where several biases are mapped to the same behavioral parameters: a future research direction is to incorporate specific biases explicitly into the model, for example, present bias through models of hyperbolic discounting, [56]. Another limitation is that the action space is small, with only two possible actions for individuals to take; an immediate avenue for future research consists of enriching the action space to include multidimensional actions (to allow for the possibility that wearing face masks may relax other protective measures), as well as heterogeneity by age and other characteristics, in such a way that the payoffs of the game depend on such characteristics. In addition, social networks could be included together in the epidemiological-behavioral model.
We view our paper as a first step in a research agenda that combines epidemiological models, game theory and behavioral economic, and we believe that it opens the possibility of incorporating interesting extensions. For example, we have included a political economy model in the Appendix, in which the government must decide how to react to the pandemic. However, there are many other economic applications that can be incorporated, among others vaccine R\&D, the vaccination campaigns, etc. We view all of these as fruitful extensions that can be incorporated into our framework.”

Reviewer 2 Report
In my view, the game-theoretic model is well grounded and executed. It has the novelty that combines epidemiological and economic variables related to COVID-19 pandemic. Likewise, it includes the role of a variety of citizens’ beliefs and perceptions in generating behaviors, which are not fully aligned with rational choice equilibria. In fact, the authors try to capture the role these beliefs and perceptions (i.e., biases) through some stylized parameters. In this sense, their proposed approach also allows for government intervention (e.g., nudging) to the aim of influencing these beliefs.
However, I would suggest that the authors better specify their research questions. In my view, these should be made more explicit, and not left to be inferred from the comprehensive reading of the text. In this manner, if the hypotheses are posed in a more explicit fashion, the contribution of the paper would be perceived more clearly.
There is minor typo in the line 587. The reference number [?] should be specified.
Author Response
Thank you for your comment. We agree that the way the paper was written, there was a lack of explicit statement of our research questions and conclusions (Referee 1 also gave us indications to improve the conclusions). We have explicitly stated our research hypothesis in the Introduction, on page 1:
“We incorporate these insights of behavioral economics into a formal epidemiological model (SEIR-like). Our research hypothesis is that including those insights would change the predictions of SEIR-like models, in relevant ways. In particular, we expect that ``behavioral interventions'' (that change the individual's perceptions of the environment) can mitigate contagion. Therefore, such epidemiological models incorporating behavioral economics can be used to study how the dynamics of the pandemic (i.e., people infected and recovered) may change due to both individual and government behavior and their interaction, since people's behavior determines to some extent the severity of the pandemic, and thus its economic consequences.”
We have also re-stated the research hypothesis in the Conclusion (on page 11), and connected it to our theoretical and empirical results:
“As stated in the Introduction, our research hypothesis was that introducing behavioral insights and game theory into a SEIR-like epidemiological model would change the predictions of those models in relevant ways. We have validated this hypothesis, by providing a series of novel results. We provide here a summary of those results.
A relevant finding is that, even if asymptomatic infected individuals are biologically less contagious than symptomatic individuals for example, due to a lower virus load, 54], they can be effectively more contagious (Proposition 1), the reason being that those with no symptoms might not take preventive measures against COVID-19 with enough intensity. However, we show that this is less likely to happen if the perceived effectiveness of protecting oneself increases (Proposition 2), as this in turn increases the perceived incentives for asymptomatic individuals to engage in protective behavior, thus reducing their infectiousness and lowering contagion. Theoretically, this means that governments can then mitigate the pandemic not only through lockdowns or by making it mandatory to wear face masks, but also by ``nudging'' people to protect themselves and the rest of society [3]}.
Concerning our numerical results, we show that an increase in the effectiveness of protective measures “flattens the curve” of contagion, as is to be expected. However, an interesting result is that, perhaps surprisingly, improving citizen's beliefs about the effectiveness of protective measures has a small effect on the pandemic dynamics. This is because at the start of the pandemic, improving citizens perceptions about protective measures has a small effect because there are few cases (and thus individual incentives are small), and by the time there are enough cases it is “too late” to meaningfully affect the dynamics of contagion. The same is true for social norms which, by themselves, might not be enough to stop the spread of the pandemic in countries in which the initial social norm is not to engage in protective behavior, for a similar reason as the one discussed for nudges about preventive measures effectiveness. For these reasons, mandates might be more effective at the beginning of the pandemic, in terms of reducing contagion, than nudges or a sudden increase in social pressure. This is one of the reasons why, in coherence with our results, mask-wearing societies with populations used to the use of protective measures have had a faster control of the pandemic [46]: the more normalized in society the use of protective measures is, the higher the probability of adopting them initially and thus when it matters most for controlling contagion and flattening the curve.
These results show that our research hypothesis was correct, and our model has produced meaningful insights that could not have been derived without considering the behavioral biases of the individuals in their decision on whether to take preventive measures.”
“There is minor typo in the line 587. The reference number [?] should be specified.”
Thank you also for informing us about this, we have included the relevant citation.
